# Surviving and Thriving: Qualitative Results from a Multi-Year, Multidimensional Intervention to Promote Well-Being among Caregivers of Adults with Dementia

**DOI:** 10.3390/ijerph18094755

**Published:** 2021-04-29

**Authors:** Meara H. Faw, India Luxton, Jennifer E. Cross, Deana Davalos

**Affiliations:** 1Department of Communication Studies, Colorado State University, Fort Collins, CO 80523, USA; 2Department of Sociology, Colorado State University, Fort Collins, CO 80523, USA; India.luxton@colostate.edu (I.L.); Jeni.cross@colostate.edu (J.E.C.); 3Department of Psychology, Colorado State University, Fort Collins, CO 80523, USA; Deana.davalos@colostate.edu

**Keywords:** caregiving, dementia, psychosocial interventions, longitudinal interventions, arts engagement, social support

## Abstract

(1) Introduction: Caring for an adult with dementia is both challenging and rewarding. Research indicates that community-based, social support, and/or arts engagement interventions can play a key role in ameliorating the negative outcomes associated with caregiving while enhancing its more positive attributes. This study explores the psychosocial outcomes experienced by dementia caregivers who participated in a multi-year, multidimensional intervention aimed at promoting caregiver and care recipient well-being. This intervention included bringing caregivers and people with Alzheimer’s disease or related dementias (ADRD) to local symphony performances, hosting a social reception prior to the performance, and assessing the outcomes of participation for both caregiver and the care recipient. (2) Materials, Methods, and Analysis: Qualitative data from participant phone interviews (*n* = 55) as well as focus groups are analyzed using thematic analysis from a phenomenological perspective. (3) Results: Across three years of participation, caregivers reported three main program benefits: relationship building (both with other participants as well as within the broader community); restored humanity (experiencing a greater sense of personal dignity and momentary return to normalcy), and positivity (experiencing positive emotions during the program). (4) Discussion: These findings point to the value of creating caregiver programming that brings together multiple dimensions of successful interventions in order to enhance caregiver experiences and positive intervention outcomes.

## 1. Introduction

More than 5.7 million Americans live with Alzheimer’s disease or related dementias (ADRD) [1], and statistics suggest that more than 16.1 million Americans provide unpaid care for an adult with dementia. This care exacts a toll among these caregivers [2], and family dementia caregivers are at an increased risk for depression, physical challenges, and burn-out [3,4]. This is especially true as ADRD progresses and individuals begin to exhibit increasingly problematic symptoms [5]. The deterioration of caregiver health has the potential to deliver a double-shot to the healthcare system. As caregiver health declines, the quality of care they can provide also declines, resulting in increased caregiving demands that exacerbate the cycle and further stress families [4].

While dementia caregiving can be taxing, evidence suggests that elements of the caregiving process can be rewarding and that providing caregivers with resources, such as social support [6] and opportunities for therapeutic arts engagement [7] can alleviate some of their perceived burdens and promote well-being. The goal of this study is to present qualitative findings from a multi-year arts engagement program offered to adults with ADRD and their caregivers, with an emphasis on understanding the specific psychosocial benefits experienced by caregivers.

### 1.1. The Cost of Alzheimer’s Disease and Related Dementias

Nearly 6 million people in the United States live with ADRD, and the number of ADRD diagnoses is projected to continue to rise [1]. Alzheimer’s disease and other dementias are neurodegenerative conditions whose symptoms progress over time until an individual is no longer able to complete basic tasks. Common dementia symptoms include memory loss, behavior changes, and difficulty speaking and walking [1]. While there is no known cure for dementia [1], its often slow-progressing nature means that diagnosed individuals can live for years after their initial diagnosis, increasingly relying on caregivers to assist them [2].

The more than 16.1 million informal dementia caregivers in the United States [2] have been referred to as the “hidden victims” of dementia [8]. Dementia caregiving is associated with a host of negative outcomes, including an increased risk of depression and physical exhaustion [3]. Evidence also suggests that ADRD caregivers experience an increased risk of developing dementia themselves [9]. While the exact pathways connecting ADRD caregiving and dementia remain unclear, researchers agree that the stress associated with caregiving likely plays a role in this pathological pathway [9].

Part of what makes ADRD caregiving challenging is that caregiving in this context results in an inherently inequitable relationship [10], with the person with ADRD dependent on the caregiver for their daily functioning but often unable to reciprocate even simple acts of love [11]. This loss of equity is difficult, as research suggests that humans value equitable relationships, choosing to minimize or manage the relationship costs they incur whenever possible [12]. In a caregiving relationship, this desire for equity goes unfulfilled, and, in some cases, caregivers report feeling guilty for their longing for a more equitable relationship, and this shame can further contribute to burnout [13,14].

While this loss of equity is problematic, dementia caregiving is further complicated by a second loss: The loss of the unique personality and shared relationship history with the person diagnosed with ADRD [15,16]. With the loss of their ability to remember and communicate, individuals can no longer “perform relationships” [17], and caregivers must negotiate their sense of grief [16]. For example, dementia caregivers often report that they have lost the “real” version of the person with ADRD. This odd paradox of loss combined with the physical presence of the person with ADRD creates an emotionally complicated environment for caregivers and contributes to an increasing sense of isolation [18].

### 1.2. Caring for Caregivers

Dementia caregiving can be challenging, and researchers increasingly recognize the importance of addressing caregivers’ needs. During the past few decades, researchers have pinpointed several avenues for enhancing caregiver health. Some programs emphasize training caregivers in their role around best practices in caregiving techniques [19,20]. Others focus on providing caregivers with enriching experiences to facilitate well-being. Among this second category of interventions, some focus on increasing caregivers’ sense of social support [21] and connection with the person with ADRD [22]. Social support includes acts to help reduce a person’s uncertainty around a challenge and help them feel connected within a broader community [23]. Across numerous studies, increased caregiver social support is associated with positive outcomes, including a decreased risk of depression [24] as well as enhanced well-being [25], life satisfaction [26], physical health [27], and satisfaction with their relationship with the person with ADRD [22].

Evidence also suggests that efforts to connect caregivers and adults with ADRD to arts engagement programming can also produce positive outcomes [28]. Extensive research on in-home and small group music therapy have illustrated the power of these experiences to lower caregivers’ levels of subjective depression [29] and burden [30], while adults with ADRD experienced reduced levels of agitation, improved mood, and some short-term memory improvements [29,31,32]. Evidence also suggests that music therapy can increase connectedness and closeness between caregivers and people with ADRD [33].

Other research has highlighted the importance of engaging in creative experiences in public spaces for caregivers and adults with ADRD. Camic and colleagues found that participants who engaged in art therapy at art galleries cited increased feelings of social inclusion that positively influenced their relationship with their person with ADRD and the broader community [7]. Similar programming involving poetry creation, photography-based storytelling, and dramatic acting found that participating in public spaces reduced perceptions of stigma around dementia for caregivers and people with ADRD [34,35]. Taken together, the evidence suggests that programs allowing caregivers and adults with ADRD to engage in enriching experiences that occur in public spaces promote a sense of belonging that can ultimately enhance well-being.

### 1.3. A Multi-Dimensional Approach to Caring for Caregivers

Given the positive outcomes resulting from interventions including these different components, it is important to consider how interventions that combine multiple elements might produce positive outcomes for caregivers and those with ADRD. These multidimensional interventions include combining multiple elements like social support, community engagement, participation in the arts, and enrolling in the intervention with the person who has ADRD [36]. Evidence from a 2003 meta-analysis suggests that caregiver interventions were most effective when social support was combined with additional intervention tasks (e.g., art activities) [37]. Additionally, this research found that caregivers were more likely to persist in interventions that included the person with ADRD, ultimately producing greater and longer-lasting benefits for caregivers [37]. Cousins et al. proposed a taxonomy of arts interventions, exploring eight unique principles (connection, engagement, expression, humanity, involvement, possibility, selfhood, and transformation), which are the elements that produce an effect for people with dementia or their caregivers [36]. When thinking about programming to address caregivers’ needs, bringing together multiple dimensions or principles can create more pronounced positive effects.

The “community music intervention” in this study was designed to incorporate several interconnecting principles, including connection, engagement, involvement, and humanity [36,38]. The present study details qualitative findings related to a multi-year intervention for adults with ADRD and their caregivers. The intervention, called B Sharp Arts Engagement^®^, promotes psychosocial well-being by providing caregivers and adults with ADRD with art engagement experiences in conjunction with a local symphony. The program was inspired by similar programs in Phoenix, Arizona (the B-Sharp Music Wellness Project) [39,40] and the Museum of Modern Art (MOMA) in New York City’s Alzheimer’s Project [41]. Past programs have focused on implementing community-based programs for people with ADRD. While programs like the MOMA Alzheimer’s Project focus on accessibility and training of arts and health professionals, the current program focused on involving people with ADRD in mainstream community activities and intentionally including caregivers in a meaningful way. Caregivers were frequently women, and caregiver relationships were adult child-parent, spousal, or, in a small number of cases, a close friend acted as the caregiver. These experiences feature opportunities for social support, engagement in a stimulating arts performance, and community connection, creating a multidimensional approached to care for caregivers and adults with ADRD. For this research, we investigate the social benefits that caregivers report from engaging in this intervention, answering the following research question: How is involvement in multidimensional programs like B Sharp associated with caregivers’ psychosocial well-being?

## 2. Materials and Methods

Data were collected as part of a larger study, and all project elements received Institutional Review Board approval (Colorado State University, protocol #15-6102H) with participants providing written informed consent. At the time of data analysis, the B Sharp program had been running for three years, with the first cohort of participants entering the program during 2015. In the program, an adult with dementia and their caregiver (usually a spouse, though adult children and friends also serve as caregivers) were provided season tickets to local symphony performances. Each symphony season included at least five concerts, with most participants attending three to five performances. Participants attended regularly scheduled symphony performances rather than attending a separate, exclusive performance. As part of the program, participants were also provided with a social reception where they could interact with other participants before each concert and during intermission. In total, each B Sharp event lasted between three to four hours. The B Sharp program was designed to function as a multidimensional intervention for ADRD caregivers and the person with ADRD. It involved both the caregiver and care recipient; it included music engagement in a public space and at a community event; and it featured a longitudinal design, with participants minimally engaging in the program over an 8-month period. In this way, it met the criteria of a multidimensional intervention by combining several interventional elements into a single program. Early evidence from the B Sharp program examining its effects on adults with dementia indicated enhanced cognitive performance [42], though data regarding caregiver experiences has not yet been assessed.

Caregiver data was collected through qualitative phone interviews after each symphony concert. These semi-structured phone interviews included questions about caregiver’s concert experiences, barriers to concert attendance, the level of social engagement experienced, etc. Additionally, all caregivers were invited to participate in annual focus groups after the end of each symphony season. During the focus groups, participants were given the opportunity to reflect on their program experiences as well as to make suggestions regarding future B Sharp events. After three years of phone interviews and focus groups, preliminary review indicated that no new information was being uncovered through these conversations, indicating data saturation. All phone interviews and focus groups were audio recorded (rather than video recorded) to preserve confidentiality and analyzed from a phenomenological tradition.

Participants were recruited through various local community organizations using snowball sampling. Staff from local nonprofit organizations (e.g., the local chapter of the Alzheimer’s Association) and other community partners (e.g., health care providers and dementia-related nonprofit organizations) identified eligible couples and invited them to join the program. To participate in the B Sharp program, one person needed to experience some form of age-related dementia-related disease. As our understanding of classifications of dementia-related pathologies expands, our focus was less on one type of dementia specifically, but the inclusion of dementia-related disorders that are accompanied by cognitive decline and loss of independence [43]. All participants needed to have at least one caregiver who was willing and able to attend symphony events with them. As this is a community-based intervention sponsored by community agencies and designed primarily as a life enrichment opportunity, no selection criteria were established for type or stage of dementia, as would be the case in a clinical trial. Recruitment occurred at the onset of each program season, with participants enrolling into the program based around a single symphony season across approximately 8 months (with concerts from October through May). Participants were not segmented in any way, and all participants were invited to engage in all intervention components.

## 3. Analysis

All audio recordings were transcribed. Transcripts were anonymized by removing all potentially identifying information, and transcripts were analyzed using thematic analysis [44]. Thematic analysis uses six distinct steps. The first step (familiarizing) involves researchers become familiar with their data. The second and third step, creating initial codes and identifying themes, involves analyzing the data for patterns and then clustering these patterns in tentative themes. The fourth step involves reviewing the themes and checking them for consistency. The fifth step includes defining and naming the themes, with the final step involving producing a written report.

In this study, familiarization was accomplished through careful reading of transcripts. Next, the researchers isolated individual utterances or, in the case of focus group data, conversational exchanges centered on caregiving experiences. After creating a set of initial codes, the researchers carefully read through these codes to refine the themes, grouping them around commonalities and assessing them for clarity and cohesion. Researchers then assessed the themes for consistency by having a separate individual familiar with the research process read and critique the themes. After making adjustments based on these recommendations, the final themes were created. In determining the final themes, the researchers relied on several key criteria, including repetition (i.e., the same ideas were present across multiple participant experiences) and forcefulness (i.e., an idea or experience was represented very strongly in participants’ discussion) as well as internal homogeneity (i.e., the experiences categorized under a single theme cohere appropriately) and external heterogeneity (i.e., each theme is conceptually distinct from the others identified in analysis) [44]. In total, the analysis process took approximately 40 hours of coding, refinement, and data engagement.

## 4. Results

Across the program’s three years, a total of 55 unique dyads (110 participants) participated in the program (for demographic information, see Table 1). Participants in the present study completed interviews and/or focus groups, with many participants engaging in both forms of data collection. As such, demographic information present in Table 1 represents the combined data for all study participants, including those who completed interviews, focus groups, or both. Nearly all participants in the B Sharp program identified as White. Caregivers were predominately female (78.18%) whereas the participants with dementia were more evenly split between men (52.72%) and women (45.45%). Across the program, participants with dementia experienced multiple forms of cognitive impairments, including Alzheimer’s disease, vascular dementia, frontotemporal dementia, unspecified dementia, Lewy Body dementia, Fragile X syndrome, and mild cognitive impairment, among others.

During its inaugural year (2015–2016), 23 dyads (46 participants) attended at least one symphony concert. During the second year (2016–2017), 26 dyads (52 participants) participated. Of these participants, 10 dyads (38.46%) had not previously participated in the program. During the third year (2017–2018), 36 dyads (71 participants, with one caregiver attending events with two individuals with ADRD) attended symphony concerts, including 22 dyads (61.11%) who were new. Across the program’s three years, 8 dyads have participated in all three years, 14 dyads have completed two years, and 33 have completed one year of the program. Participants who declined to participate in the B Sharp program for more than one year cited several reasons for discontinuing the program, including the institutionalization or death of the person with ADRD, loss of interest in the symphony, or the progression of the person with ADRD’s condition to a point where attending regular events became too difficult. Results yielded three key themes: relationship building, restoring humanity, and positivity (see Table 2).

### 4.1. Building Relationships

The most prominent theme that emerged from the data was the importance of B Sharp for building relationships. In total, 61.81% of participants (*n* = 34) discussed the value and importance of building relationships as an essential benefit for the B Sharp program. For example, one participant (female spouse, age unknown) highlighted the rarity with which she was able to engage in social experiences and how B Sharp provided her the opportunity to connect with others:
I think just the fact that we got out of the house! On the [research] questionnaire, it said, “How often do we go to a social event?” And we never hardly get to go anywhere. So it was great to be out among other people […]

Through their symphony attendance, participants discussed how important it felt to connect with new people. As one participant (female spouse in her 70’s) explained:
To me, it’s like a mini support group […] It’s fascinating to me to meet other people and caregivers. […] It helps that I know there are others, and we can learn things from each other […] It’s like a club […] just in that one room.

During a phone interview, another participant (female spouse in her 50’s) talked about the program as providing nourishment to help her be a better caregiver. By connecting with others through B Sharp, she felt greater capacity to take on the challenges of caregiving:
I find connecting with other caregivers at that time […] It’s almost more as if we have all been fed. So we’re coming from a place of strength. And I notice a difference in the way the caregivers talk, and it would be different for me as well […] there’s just a sense of purpose. It’s almost as if you’re not lost in your caregiving role […] I just find that, overall, there’s a strength between the caregivers.

Along these same lines, participants also talked about the power of bonding over caring for a person with ADRD for building resilience. As two participants (both female) discussed during a focus group:
Person C: It’s scary when you see how many people are in your focus group.
Person D: It’s an ongoing thing, and it’s getting worse.
Person C: But I think we help each other be stronger. I think it’s a community. I have a lot of friends we do stuff with […] But they don’t really know what it’s like. They sympathize. They know there’s something different, but nobody really realizes until you get with people that are living it.

In this way, participants recognized the value of making both interpersonal and communal connections through B Sharp. Many participants identified this sense of solidarity as an essential B Sharp benefit. These participants found power in being with others who inherently understood their experiences. One participant (female spouse in her 70’s) explained the freedom she experienced from knowing that others in B Sharp understood her situation, like not having to explain her husband’s fidgeting during the concert:
I feel more comfortable amongst people who know what you’re going through, and you don’t have to explain if your person does something inappropriate or whatever. You have to be ready as a caregiver, and it’s easier to not feel embarrassed or bad about it. It’s just the way it is. You are just more at ease. Even though we all come from different backgrounds, inside of that room, it’s just—you realize that it can hit anyone, and we are all dealing with it.

For some participants, the new relationships formed through B Sharp moved beyond interpersonal connections and constituted a new community or a renewed connection to the community. Within the focus groups, participants chiefly talked about the power of B Sharp to help them feel a sense of belonging and acceptance within the broader community. Being able to talk with and see others who shared similar experiences helped these participants to feel less alone. As two participants (both female) discussed during a focus group:
Person A: We moved here from Florida because so many of our friends were gone, and my daughter says, “Why not come out here? This is a great town”. Which I knew. I had been out here many times. I was stunned by the response that [the community] gives to people with Alzheimer’s and dementia. I’ve never seen such caring from a town.
Person B: That was a big thing I think that the program did. I think it made us feel like people were noticing us and caring about us, doing something for us […]

For these caregivers, seeing how the community recognized and responded to B Sharp participants helped alleviate their sense of isolation. As one participant (female spouse in her 70’s) explained:
They mention [B Sharp] in the [symphony] program, and at every concert. And I think, “Someone cares about us”. Someone is trying to help us. It’s a feeling of someone doing something for this disease and trying to solve it […] It’s been a nice thing. It’s like someone giving me a gift. It’s like a present.

### 4.2. Restoring Dignity and Humanity

Participants also highlighted the importance of B Sharp for restoring a sense of dignity and humanity. This was the second most common theme emerging from the data, with 27.27% (*n* = 15) indicating that restored dignity and humanity were key benefits of the B Sharp program. For participants in the focus group, the sense of dignity came not just for themselves but also for their partner, underscoring the value of B Sharp’s intervention design featuring both caregivers and people with ADRD enjoying community events together. As one participant (female spouse, age unknown) explained to the group:
Do you know what’s different [about B Sharp], the greatest thing is? When you come into a group like this, you’ll come up to my husband and say, “Hi [Husband], you’re looking good”, something like this right there. You go to church, and they’ll come up, “Hi, [to me]”. They’ll look at your husband, and walk away. I mean don’t you, do you notice that?

In the focus groups, participants also voiced their opinions about how B Sharp made them feel a greater sense of humanity through recognition and providing the opportunity to participate in a dementia-friendly community event. The formality of the event also helped provide a sense of dignity. As one participant (female spouse, age unknown) explained in a focus group when responding to another participant’s observations about the formality of the symphony:
This is just very nice, kind of like you were saying at the beginning. You said, “It’s a little too high [class]”, or something. In a sense, it gives dignity, I feel, to both of us—or to all of us. To me, it’s a nice way of having something to do together when you feel like you’ve lost all the things you used to be able either to do together or separately.

Other caregivers felt a sense of dignity through the ability to look forward to something. As the prognosis for ADRD can be bleak, many participants enjoyed the emotional escape of anticipating a symphony concert as a break from their daily living. As one participant (female spouse, age unknown) articulated in her interview, “It gives me a break. It’s something I look forward to. It’s on the calendar. I love getting out and having a way to feel like myself”. The breaking of routine felt like a powerful source of dignity, as another participant (female spouse, age unknown) explained, “[…] It’s like a break from our regular life. It feels light-hearted. Even the snacks there—I don’t have to worry about food! I get to relax and play a little bit. It feels good”.

For others, a sense of humanity was restored in being able to explore their identity outside of the caregiver role. For example, one participant expressed her appreciation at being able to discuss different topics at B Sharp events, “It’s a more normal social setting. I get to chat about things such as politics with others […]”. Another participant highlighted the value of being able to temporarily suspend her worries, “It makes you feel like you are involved in the world again, not worrying about other things”. Another participant (female spouse in her 70’s) acknowledged the value she found from participating in this space that, while designed to serve those dealing with dementia, allowed caregivers to put aside this role, “Just having something to do and get out of the house and be around different people. It takes a different place, emotionally and mentally, from the dementia”. For these participants, a benefit of B Sharp was the ability to feel a sense of renewed humanity and dignity as individuals, not just as constant caregivers.

When discussing their caregiving role, participants often highlighted the loss of normalcy in their relationship with their partner. Attending B Sharp events helped them feel more normal. As one participant (female spouse, age unknown) explained, “[B Sharp] gives us all a night to dress up and play normal adults. There’s a lot of child-parenting going on, and then when you’ve got your clothes on and show up and mingle with other adults, it feels normal again”. Another participant (female spouse in her 70’s) echoed this sentiment, highlighting how the events allowed caregivers a reprieve from “parenting” their partner: “It’s a grown-up thing to do and not a child-parent thing […] That’s always a rewarding feeling when we go out in public together. A concert is a normal, adult thing to do in the context of Alzheimer’s progression”.

### 4.3. Positivity

A final theme that participants’ interviews highlighted regarding the value of B Sharp was its ability to foster positivity, with 10.90% (*n* = 6) participants specifically calling out positivity as a central component of the B Sharp experience.

In discussing their experiences with B Sharp, many participants acknowledged the seriousness of their partner’s prognosis. However, several participants also felt overwhelmed at the perceived negativity around ADRD conversations. These participants recognized that B Sharp, unlike other events, focused on creating positivity. As one participant (female spouse in her 80’s) discussed: “It’s just having people around in an upbeat mood. There’s a lot more negativity in support groups. I’m more excited about the concerts and those interactions than the support groups […] I like talking to the other caregivers. Everyone is upbeat and not focusing on their problems”.

Another participant (female spouse in her 60’s) explained that her experiences in support groups left her feeling depressed. With B Sharp, though, she found positivity, “I look forward to these performances because they bring me a lot of peace. Everyone seems happier and interested in the conversations from our past, which are really fun to have”. Reflecting on the performances, one participant (female spouse in her 70’s) commented, “[B Sharp] just makes you happy. Everyone leaves happy. If you watch everybody leaving, they’re all smiling. Nobody’s with a gloomy face”. Another caregiver (female spouse, age unknown) echoed this, explaining the personal sense of fulfillment she received from participating in a positive event: “All I can tell you is that it still tickles my heart […] It felt like I was pampered […] It felt very special and nice […] For me, it’s joy. It felt kind of freeing and special”.

## 5. Discussion

Dementia caregivers can experience immense challenges in their role, leading to scholarly recognition that informal caregivers are the “hidden victims” of dementia [8]. The B Sharp program was specifically designed to address some of the complex needs of caregivers and people with ADRD by providing them with tickets to symphony concerts featuring pre-event receptions that emphasize the importance of inclusion and multiple social connections. Follow-up phone call interviews as well as data from end-of-year focus groups across three years of programming suggest that participation in B Sharp results in positive experiences for caregivers, including relationship building, an increased sense of dignity and humanity, and a sense of positivity.

These results generally support claims that caregivers need opportunities to engage with others, and that these opportunities can be limited because of their role. When discussing the benefits of B Sharp, participants acknowledged the power of relationships built through B Sharp across multiple levels—from interpersonal connections to a sense of belonging in the larger community. Previous research has illustrated that caregiving is often isolating [18], and addressing isolation is essential to promoting caregiver well-being [45]. These results illustrate that loneliness can and should be addressed at multiple levels. For some participants, building one-on-one relationships through B Sharp resulted in greater feelings of support. Other participants, however, benefited from simply attending normal events alongside other community members. While this second form of connectedness may not carry the same resources that close, supportive relationships can offer [23], it may prove valuable in reducing feelings of stigma, which are often associated with increased risk of depression among caregivers [14]. Future research should continue to explore the different levels of social connectedness (e.g., at the interpersonal level versus the community level) to identify the unique benefits conferred by each for caregivers of adults with ADRD.

Interestingly, while participants identified relationship building as an important component of B Sharp, they also talked about the B Sharp program’s positivity, with a few participants going so far as to contrast the positivity of B Sharp with the perceived negativity of support groups. While research has consistently found support group participation generally helpful in promoting caregiver well-being [46], some participants felt that their support group experiences focused too strongly on negative caregiving experiences. For these participants, B Sharp offered a refreshing contrast, with participants experiencing a lighter side of life through the events. These experiences introduce an interesting question regarding the power of social support: When does social support promote positive outcomes, and when might it promote negative outcomes? What types of events are better at promoting both positive emotions and positive relationships? The theory of human well-being identifies our need for both positive relationships and positive emotions [47], yet research on social support has found negative effects associated with co-rumination and highlights the potential for supportive conversations to veer into unhealthy territory [48]. Future research should explore when and how co-rumination might occur in caregiving support groups as well as strategies to accomplish balance between troubles talk with more positive forms of sharing.

Another important B Sharp benefit was a sense of restored dignity and humanity. Part of this came from being able to do something normal with the person with ADRD. Many participants voiced nostalgia for previous social experiences, and going to a concert as a couple helped caregivers temporarily step out of their caregiver role and connect with their partner on a more equitable footing. As caregiving relationships are inherently inequitable [10], having the opportunity to restore a sense of equity, even briefly, felt meaningful to caregivers. This echoes previous findings about the importance for couples dealing with dementia to engage in events together that do not center solely around dementia treatment [37]. Providing outlets for a greater sense of equity could result in enhanced relationship quality, which, in turn, could provide important benefits to both relationship partners [49]. Future research should continue to identify other experiences that might help restore equity (even temporarily) in the caregiver-care recipient relationship.

Taken together, this study highlights the importance of understanding caregivers’ experiences. While caregivers have received increasing empirical attention, much dementia programming continues to focus primarily on the person with dementia, with fewer efforts made to understand caregiver needs and the best ways to provide them with systematic care [50]. Given the inextricable link between caregiver health and care recipient health [4], it is essential that researchers and practitioners continue to work to identify the most effective ways to address caregiver needs.

While the findings from this study point to promising future research, it is not without limitations. The participants in this study all come from one geographic area, Northern Colorado. As such, it is a fairly homogenous group. As evidence suggests that caregivers can experience outcomes differently across racial [6], gender [24], and cultural lines [51], it is important to investigate interventions and their effects across diverse populations. Additionally, we did not segment participants based on their diagnosis, years of caregiving, or other demographic features, which limits our ability to speak about B Sharp’s interventional effectiveness across diverse caregiving circumstances. We also gathered data through audio recordings only. While this method was chosen in order to protect participants’ confidentiality, the loss of important nonverbal cues is also a limitation of this work. Finally, though all B Sharp caregivers were invited to participate in phone interviews and focus groups, it is likely that those who had positive experiences were more motivated to be involved. As such, our results might reflect a positivity bias. In the future, greater efforts to connect with participants who report a broad range of reactions to interventions will be important to identify the benefits and potential drawbacks of programming like B Sharp.

Assessments of community-initiated interventions have both strengths and weaknesses. First, they provide the opportunity to study genuine social events and their outcomes because they are sponsored and hosted by community agencies rather than simulated lab experiences. Thus, the results have strong external validity and the assurance that the program outcomes are the result of a genuine social context and not an artifact of the research experience. A second strength is their reproducibility; as a community-sponsored event, they provide a model for adoption in other community contexts. These same strengths also create limitations. First, community interventions recruit interested parties, and therefore may be prone to selection bias, only demonstrating the potential for parties interested in community engagement events, not all pairs facing dementia and caregiving struggles. Second, they do not select participants based on stage or type of dementia. This inclusion of participants with varying types and stages of dementia does not allow for a robust assessment of differences in outcomes or experiences across different types of dementia. Third, the sample is necessarily limited by the size of the community, number of interested parties, and ability of the sponsoring agencies to recruit participants. To understand the broader potential of programs like this would require coordination across multiple sites to gather data from multiple contexts and a larger sample.

## 6. Conclusions

Despite its limitations, this research highlights important, practical findings for those wishing to provide care for ADRD caregivers, such as the importance of creating opportunities where caregivers and adults with ADRD can engage with their community. This visibility serves multiple ends: It helps caregivers feel less stigma and isolation while also providing more visibility and awareness of dementia [34]. Findings from this research also underscore the importance of feeling normal and experiencing positive interactions—things that can be achieved by bringing caregivers and people with ADRD together in ways that balance recognizing the role of ADRD in their lives without overemphasizing it. The sense of feeling seen and welcomed in the community is a key insight for how community-based interventions can support people who otherwise feel forgotten or unseen. In this case, it was not just participation in the program that led to that greater sense of being cared for, but the public mention of B Sharp program at the beginning of each symphony performance which created a positive impact for caregivers. Finally, this study highlights the importance of implementing programming that features multiple dimensions (e.g., social connection, creative engagement, and public presence) to address the complex needs of caregivers and adults with ADRD.

## Figures and Tables

**Table 1 ijerph-18-04755-t001:** B Sharp Participant Demographic Information.

**Sex of Person with Dementia**	**%**	*N*
Male	52.72%	29
Female	45.45%	25
Unknown	1.81%	1
**Sex of Caregiver**	**%**	*N*
Male	21.81%	12
Female	78.18%	43
Unknown	0.00%	0
**Relationship to Person with Dementia**	**%**	*N*
Spouse/Partner	67.27%	37
Child or Child-in-Law	23.64%	13
Friend	3.64%	2
Professional Caregiver	3.64%	2
Unknown/Not reported	1.81%	1
**Age of Participant with Dementia**	**%**	*N*
50’s	3.63%	2
60’s	9.09%	5
70’s	14.54%	8
80’s	30.91%	17
90’s	5.45%	3
Unknown/Not reported	36.36%	20
**Age of Caregiver**	**%**	*N*
30’s	1.81%	1
40’s	1.81%	1
50’s	10.90%	6
60’s	18.18%	10
70’s	9.09%	5
80’s	7.27%	4
Unknown/Not reported	50.09%	28
**Type of Dementia/Cognitive Impairment**	**%**	*N*
Alzheimer’s Disease (including early onset)	41.81%	23
Unspecified Dementia	12.72%	7
Frontotemporal Dementia	1.81%	1
Vascular Dementia	9.09%	5
Fragile X Syndrome	1.81%	1
Mild Cognitive Impairment	16.36%	9
Lewy Body Dementia with Parkinson’s	1.81%	1
Unknown/Not reported	14.65%	8

Note: All demographics represent the combined reports across all three years of participants. Unknown data is due to missing information collected during the intake process, such as participants neglecting to provide information on date of birth, sex, or diagnosis in their demographic profile. In some cases, demographic information was gathered over the phone and if a participant was unable to be reached, we could not gather this information.

**Table 2 ijerph-18-04755-t002:** Summary of Thematic Results.

Theme	Prevalence	Definition	Exemplar Quotes
Relationship building	61.81% (*n* = 34) Interviews & focus groups	Increased feelings of connection with other B Sharp participants and with the broader community	To me, it’s like a mini support group […] It’s fascinating to me to meet other people and caregivers. […] It helps that I know there are others, and we can learn things from each other […] It’s like a club […] just in that one room. (*Female spouse in her 70’s*) They mention [B Sharp] in the [symphony] program, and at every concert. And I think, “Someone cares about us”. Someone is trying to help us. It’s a feeling of someone doing something for this disease and trying to solve it […] It’s been a nice thing. It’s like someone giving me a gift. It’s like a present. (*Female spouse in her 70’s*)
Restored dignity and humanity	27.27% (*n* = 15) Interviews & focus groups	Experiencing increased recognition of their life outside of caregiving as well as greater respect for the person with Alzheimer’s disease or related dementias (ADRD)	It’s a grown-up thing to do and not a child-parent thing […]. That’s always a rewarding feeling when we go out in public together. A concert is a normal, adult thing to do in the context of Alzheimer’s progression. (*Female spouse in her 70’s*) Just having something to do and get out of the house and be around different people. It takes a different place, emotionally and mentally, from the dementia. (*Female spouse in her 50’s*)
Positivity	10.90% (*n* = 6) Interviews	Benefiting from the general positivity present at B Sharp events	[B Sharp] just makes you happy. Everyone leaves happy. If you watch everybody leaving, they’re all smiling. Nobody’s with a gloomy face. (*Female spouse in her 70’s*) All I can tell you is that it still tickles my heart […] It felt like I was pampered […] It felt very special and nice […] For me, it’s joy. It felt kind of freeing and special. (*Female spouse, age unknown*)

## Data Availability

The data presented in this study are available on request from the corresponding author. The data are not publicly available due to privacy concerns and restrictions from the Institutional Review Board.

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
