# Peer review of "Surviving and Thriving: Qualitative Results from a Multi-Year, Multidimensional Intervention to Promote Well-Being among Caregivers of Adults with Dementia"

_ijerph, 2021, doi:10.3390/ijerph18094755_

Round 1

Reviewer 1 Report

The manuscript brings interesting findings about an art intervention in well-being of dementia/ cognitive decline caregivers and presents high level of merit in terms of care, mental health and humanized treatment. However, I have suggestions regarding to the Title, Abstract and Body manuscript.

Title: Why the author considered the intervention as multidimensional? I think that is necessary to conceptualize what is multidimensional intervention in the “Introduction” section, as well, describe what dimension the B Sharp program included in “Material and methods”.  

Abstract: The abstract is poor in terms of all work made in the study. I suggest to short the first sentence and described briefly what consisted the intervention.

Body manuscript:

Which is the profile of caregivers? Generally, Who assumes the care? Are they women, children, spouse…? Please clarify in the Introduction.

There is a “s” letter in the middle of sentence located in line 118. Please correct.

In the Introduction, please provide more details regarding to the B Sharp Arts Engagement® and the evidence described in previous study. The information in lines 141-143 are more adequate in Introduction than “Methods”.

The number of participants in the Table 1 and line 159 are inconsistent.

During the symphony performances, only caregivers and patients attending? It is not clear if they received the ticket in a regular session or if the session was exclusive to the study participants.

What are the reasons to unknown sex and age? Please inform.

I miss more details about participants characteristics. Please describe the proportion of participants among the diseases AD, Mild cognitive impairment, Levy dementia and others. What were the relationship between the caregiver and patient? Were they children, spouses, friends? In which proportion? Please present this information in Table 1.

The number and frequency of found themes should be described. The “Building Relationships”, “Restoring Dignity and Humanity” and “Positivity” occurred with higher frequency? The criteria to define the three major results is need.

Standard the results presentation related to the speech of caregivers, following the format in the subheading 3.1. Please see lines 283-285 and 287-290.

The discussion is well-written and interesting.

Line 427: the study included only AD caregivers?

Author Response

Thank you for your thoughtful review of our manuscript. Below, we have provided a point-by-point response to each of your comments. Our comments are bolded and italicized.

The manuscript brings interesting findings about an art intervention in well-being of dementia/ cognitive decline caregivers and presents high level of merit in terms of care, mental health and humanized treatment. However, I have suggestions regarding to the Title, Abstract and Body manuscript.

Title: Why the author considered the intervention as multidimensional? I think that is necessary to conceptualize what is multidimensional intervention in the “Introduction” section, as well, describe what dimension the B Sharp program included in “Material and methods”.  

We have added additional information regarding the multidimensional components of B Sharp. First, we updated the abstract to highlight the different components of the B Sharp program (per your comment related to the abstract; see the next bullet point). Then, we more explicitly define “multidimensional” on p. 3, lines 137-139. We have also included a clarifying statement about the multidimensionality of B Sharp on p. 4, lines 211-217.

Abstract: The abstract is poor in terms of all work made in the study. I suggest to short the first sentence and described briefly what consisted the intervention.

The abstract has been modified to include a brief description of the intervention.

Body manuscript:

Which is the profile of caregivers? Generally, Who assumes the care? Are they women, children, spouse…? Please clarify in the Introduction.

Additional information about the profile of caregivers has been added in the introduction (see pg. 3, 163-165).

There is a “s” letter in the middle of sentence located in line 118. Please correct.

Thank you for pointing this out. We have removed the errant “s”.

In the Introduction, please provide more details regarding to the B Sharp Arts Engagement® and the evidence described in previous study. The information in lines 141-143 are more adequate in Introduction than “Methods”.

Thank you, the text has been moved from Methods to the Introduction and expanded upon (see pg. 3, 156-165).

The number of participants in the Table 2 and line 159 are inconsistent.

We have corrected this error. Table 1 represents the demographic information for 55 people with dementia and 55 caregivers. In line 351, we have updated this to reflect 110 participants instead of 109.

During the symphony performances, only caregivers and patients attending? It is not clear if they received the ticket in a regular session or if the session was exclusive to the study participants.

We have added additional information to clarify that caregivers and participants attended the regular symphony sessions, rather than an exclusive performance (see pg. 4, lines 207-209).

What are the reasons to unknown sex and age? Please inform.

We have added additional information on this as note in Table 2. Unknown data is due to missing information collected during the intake process, such as participants neglecting to provide information on date of birth and sex in their demographic profile. In some cases, demographic information was gathered over the phone and if a participant was unable to be reached, we could not gather this information. 

I miss more details about participants characteristics. Please describe the proportion of participants among the diseases AD, Mild cognitive impairment, Levy dementia and others. What were the relationship between the caregiver and patient? Were they children, spouses, friends? In which proportion? Please present this information in Table 1.

We have updated this table (now Table 2) to include more demographic information, including the type and proportion of diagnoses as well as the nature of the relationship between the person with ADRD and the caregiver.

The number and frequency of found themes should be described. The “Building Relationships”, “Restoring Dignity and Humanity” and “Positivity” occurred with higher frequency? The criteria to define the three major results is need.

We have updated the results to indicate the frequency with which participants reported each theme (see pg. 8, lines 375-377; pg. 9, lines 461-463; and pg. 10, lines 507-509). We have also included a statement outlining the criteria by which each theme was derived (see pg. 5, lines 338-344).

Standard the results presentation related to the speech of caregivers, following the format in the subheading 3.1. Please see lines 283-285 and 287-290.

We have adjusted this quotes to present them in a block (rather than in-text).

The discussion is well-written and interesting.

Thank you!

Line 427: the study included only AD caregivers?

We have updated the language throughout the manuscript to reflect the diversity of diagnoses among our participants. Instead of referring to our caregivers as Alzheimer’s disease (AD) caregivers, we have opted to use the more inclusive term Alzheimer’s disease and related dementias (ADRD) throughout the manuscript.

Reviewer 2 Report

This study explores the psychosocial outcomes experienced by dementia caregivers who participated in a 3-year intervention of art events.  Qualitative data from participant phone interviews as well as focus groups are analyzed using thematic analysis. While there are merits in the study, some limitations and shortcomings have to be addressed.

  1. As the authors mentioned in the manuscript,  there are similar programs in Phoenix and New York City, and probably more. A more comprehensive study including different cohorts would be meaningful for this kind of study. This present study includes only one community, which is flimsy.
  2. On page 4, line 152, the sentence states that participants are from various community organizations. But in line 405, the participants are only from one community. Actually, the details of the community are not provided, not even the location.
  3. The details of the events provided by B sharp should be further elaborated, such as frequency, participating rates, etc. Comparisons of different types of art interventions are of interest. 
  4. Table 1 provides the information of the participants. There are age information of the participants with dementia, but not those of the caregivers. The numbers of participants with dementia and the caregivers do not match. 

Author Response

Thank you for your thoughtful review of our manuscript. Below, we have provided a point-by-point response to each of your comments. Our comments are bolded and italicized.

This study explores the psychosocial outcomes experienced by dementia caregivers who participated in a 3-year intervention of art events.  Qualitative data from participant phone interviews as well as focus groups are analyzed using thematic analysis. While there are merits in the study, some limitations and shortcomings have to be addressed.

As the authors mentioned in the manuscript,  there are similar programs in Phoenix and New York City, and probably more. A more comprehensive study including different cohorts would be meaningful for this kind of study. This present study includes only one community, which is flimsy.

Unfortunately, we do not have access to data from these other programs. While these programs are similar to the one featured in this manuscript, they do not include the key research component like ours does. So while we agree with this reviewer’s observation (having data from more than one cohort across multiple data collection sites would make for a powerful study), we are unable to incorporate any data from the programs offered at these other sites.

On page 4, line 152, the sentence states that participants are from various community organizations. But in line 405, the participants are only from one community. Actually, the details of the community are not provided, not even the location.

We have clarified line 405 to reflect that participants are from one geographic location. Participants were recruited from various community organizations, but all live in the same geographic area.

The details of the events provided by B sharp should be further elaborated, such as frequency, participating rates, etc. Comparisons of different types of art interventions are of interest. 

We have expanded on details about the B Sharp program (number of concerts, frequency of participation, months of events, hours per event) in the Methods and have added a comparison to other programs in the Introduction (see pg. 3, lines 151-167; pg. 4, 234-246).

Table 1 provides the information of the participants. There are age information of the participants with dementia, but not those of the caregivers. The numbers of participants with dementia and the caregivers do not match. 

We have updated Table 1 (now Table 2 in the manuscript) to include additional information about the caregivers (including age and relationship to the person with ADRD) as well as the type of diagnosis and diagnosis frequency among participants with ADRD. We also went back and double-checked the numbers. All numbers should match and reflect accurate information.

Reviewer 3 Report

Interesting research that addresses an original and impactful topic, such as the survival of caregivers of people with dementia, from a qualitative perspective.
I have really enjoyed reading this article and I think it has a lot of potential. However, I think that some important improvements need to be addressed in order to be of interest to the entire scientific community:
The summary would be much more understandable if it were structured by headings. The summary should already specify whether the qualitative approach has been phenomenological or ethnographic.
In the introduction there is something confusing, since a whole theoretical framework based on Alzheimer's and its caregivers is developed and, however, the research work presented seems to extend to caregivers of all types of dementia. This must be clarified as the approach is qualitatively different.
A greater and better description of the B Sharp Arts Engagement technique, which seems to be the focus of the planned investigation, is missing in the introduction.
When previous works that address interventions in caregivers of patients with Alzheimer's are presented, there are fewer references regarding reviews made on the available literature related to the training of caregivers, such as: 

Guerra-Martín MD, Rufino-Núñez MB, Ponce-Blandón JA, Mendes-Lipinski J. Training of informal caregivers and quality of care. A systemac review. Saúde em Redes. 2018; 4(3):101-113. http://dx.doi.org/10.18310/2446-4813.2018v4n3p101-113

In the methodology: the type of qualitative design (phenomenological, ethnographic, etc.) is not described or justified. Nor are the procedures for selecting the participants correctly described and whether the participants were segmented based on any of the sociodemographic variables collected . It is not stated what criterion was used to identify the saturation of the information, and the discourse analysis procedure must be more detailed.
The results should include the data related to the characterization of the sample included in the methodology section. In the identification of the three main thematic areas, regarding the quotations of the participants, they should be presented in a better order (for example, through tables with the quotations and characteristics of the participant mentioned).
In the discussion and in the conclusions, there are some shortcomings, such as a greater description, recognition and discussion of the limitations, beyond those already described, including some of those that have been suggested in this review. I also recommend distinguishing in the discussion between dementia in adults in general and Alzheimer's, terms that are often confused.

Author Response

Thank you for your thoughtful review of our manuscript. Below, we have provided a point-by-point response to each of your comments. Our comments are bolded and italicized.

Interesting research that addresses an original and impactful topic, such as the survival of caregivers of people with dementia, from a qualitative perspective.
I have really enjoyed reading this article and I think it has a lot of potential. However, I think that some important improvements need to be addressed in order to be of interest to the entire scientific community:

The summary would be much more understandable if it were structured by headings.

We have adjusted the abstract to incorporate the following headings: Introduction; Materials, Methods, and Analysis; Results; and Discussion.

The summary should already specify whether the qualitative approach has been phenomenological or ethnographic.

We have updated the summary to address that the results represent a phenomenological approach to qualitative analysis.

In the introduction there is something confusing, since a whole theoretical framework based on Alzheimer's and its caregivers is developed and, however, the research work presented seems to extend to caregivers of all types of dementia. This must be clarified as the approach is qualitatively different.

We have addressed how the inclusion of participants was based on each participant having a diagnosis of a type of dementia that is characterized by cognitive deficits and loss of independence.  The requirement of these two characteristics should still fit within the theoretical framework described.  In addition, a citation has been added regarding how are classification and diagnoses of dementia-related disorders is ever evolving and should be considered when we try to only include participants with one type of dementia as it is likely that many individuals with pathological aging are diagnosed with Alzheimer’s disease, and later found to have a different type of dementia.

A greater and better description of the B Sharp Arts Engagement technique, which seems to be the focus of the planned investigation, is missing in the introduction.

We have expanded on our description of the B Sharps Arts Engagement program in the Introduction and the Methods.

When previous works that address interventions in caregivers of patients with Alzheimer's are presented, there are fewer references regarding reviews made on the available literature related to the training of caregivers, such as: Guerra-Martín MD, Rufino-Núñez MB, Ponce-Blandón JA, Mendes-Lipinski J. Training of informal caregivers and quality of care. A systemac review. Saúde em Redes. 2018; 4(3):101-113.

We have added a brief discussion of these types of interventions into the introduction (see pg. 95-99).

In the methodology: the type of qualitative design (phenomenological, ethnographic, etc.) is not described or justified. Nor are the procedures for selecting the participants correctly described and whether the participants were segmented based on any of the sociodemographic variables collected . It is not stated what criterion was used to identify the saturation of the information, and the discourse analysis procedure must be more detailed.

The manuscript has been updated to indicate our use of a phenomenological qualitative design (see pg. 4, line 229) as well as the addition in the abstract. We have updated the manuscript to include information on participant inclusion criteria (pg. 4, 243-242) as well as a direct statement addressing the fact that participants were not segmented in this study (pg. 4, lines 244-246). We have included information regarding data saturation (pg. 4, lines, 226-228). We have added some additional information regarding our analytical strategy in response to other reviewers’ comments (see pg. 5, lines 338-344). However, we will note that we did not conduct a discourse analysis in the manuscript; rather, our results come from a thematic analysis—which is different from discourse analysis in that it focuses on key words or ideas that emerge from the text whereas discourse analysis focuses on discursive techniques present in the text (see https://discoursevsthematic.wordpress.com/2015/11/11/discourse-steps/) . These differences can be subtle, but are important. Because we did not complete a discourse analysis, we have not added additional information to the manuscript pertinent to discourse analysis methods.

The results should include the data related to the characterization of the sample included in the methodology section. In the identification of the three main thematic areas, regarding the quotations of the participants, they should be presented in a better order (for example, through tables with the quotations and characteristics of the participant mentioned).

Per your suggestion, we have moved the description of our participants from the “Materials and Method” section to the beginning of the “Results” section (see pg. 6). We have also added a new table summarizing the results with each theme, its prevalence, definition and exemplar quotes to illustrate the theme (see Table 1). We have also gone through the data in the “Results” section and added participant information (gender, relationship to person with ADRD, and age) throughout.

In the discussion and in the conclusions, there are some shortcomings, such as a greater description, recognition and discussion of the limitations, beyond those already described, including some of those that have been suggested in this review. I also recommend distinguishing in the discussion between dementia in adults in general and Alzheimer's, terms that are often confused.

We have worked to address this suggestion by adding additional comments related to avenues for future investigations (see pg. 11, lines 563-470) as well as a deeper discussion of the study’s limitations (pg. 12, lines 612-629).

Round 2

Reviewer 2 Report

This manuscript presents an interesting and potentially important method to promote caregivers' well-being. However, the background of the respondents is limited to one program. When the research can access many respondents, one would expect quantitative analyses also to be included. For qualitative research, more depth and width in respondents' opinions are expected, which might require respondents to have exposure to diversified and various experiences. This research is fundamentally lacking the elements in the beginning. Thus, the results cannot represent ones from an appropriately designed study.  In my opinion, the quality of this study does not meet the standard of IJERPH. 

Author Response

Thank you for this review.

Reviewer 3 Report

The manuscript has been improved in a very clear way, so authors can be congratulated for the effort. However, there are still some suggestions for improvement that I dare to make:   - The heading of the summary called: Material, methods and analysis is sufficient that it is called "Material and methods"   - I believe that table 1 corresponds to results and not to the section on material and methods / analysis   - If the two main procedures for collecting information are telephone interviews and focus groups, I think that the results that come from telephone interviews should be better differentiated from those that come from focus groups.   - Also in the demographic description of Table 2, the participants in the telephone interviews and those in the focus groups could be differentiated   - I miss a greater detail of the ethical cautions. Was there informed consent of the participants? What steps were taken to preserve confidentiality? Was there an ethical research committee that approved the procedure?   - In the discussion I also miss a further analysis of the limitations of the study. For example, the non-segmentation of the participants is recognized but this may have some effect on the results. Or also, the interviews were by telephone, so a lot of non-verbal information is lost that could be interesting for the analysis

Author Response

Comments and Suggestions for Authors

The manuscript has been improved in a very clear way, so authors can be congratulated for the effort. However, there are still some suggestions for improvement that I dare to make:  

Thank you for your kind words and your continued suggestions on this manuscript. We have responded to your comments below and believe that the manuscript is strong as a result.

- The heading of the summary called: Material, methods and analysis is sufficient that it is called "Material and methods"  

We have updated this heading to reflect your suggestion.

- I believe that table 1 corresponds to results and not to the section on material and methods / analysis  

We had placed Table 1 in its current location based on the request of a different reviewer in our first round of revisions. We have now moved this table (now Table 2 in the revised manuscript) to be a part of the results.

  - If the two main procedures for collecting information are telephone interviews and focus groups, I think that the results that come from telephone interviews should be better differentiated from those that come from focus groups.  

We have worked to further differentiate the results from each theme that come from the interviews vs. the focus groups. Now, all exemplar quotes are lumped together by data collection modality. On pg. 8-9, the results for the focus group respondents that correspond the “Building Relationships” theme are all together. Similarly, all focus group findings for the “Restoring Dignity and Humanity” theme are placed together on pg. 10. We have also clarified that the “Positivity” theme results came strictly from the interviews (pg. 10) and have made note of this in Table 2 as well.

- Also in the demographic description of Table 2, the participants in the telephone interviews and those in the focus groups could be differentiated  

All participants (those who completed the phone interviews and the focus groups) are the same participants. Thus, the demographic information represented in Table 1 is true for both focus group participants and interview participants. We have worked to better clarify this in the manuscript (see pg. 4, lines 229-233); however, given that participants are the same, we did not think it made sense to present the same demographic results for participants in two distinct tables. If the reviewer and editor feel strongly about this, we can go back and re-create Table 2 again, clarifying that one table represents the participants from the interviews and the other from the focus groups (though, again, those tables will have the exact same information).

- I miss a greater detail of the ethical cautions. Was there informed consent of the participants? What steps were taken to preserve confidentiality? Was there an ethical research committee that approved the procedure?

We state on pg. 4, lines 154-156 that all procedures were approved by an Institutional Review Board and that all participants (caregivers and PWD alike) provided written informed consent. We have sought to clarify that participants’ confidentiality was preserved through audio recording interviews/focus groups (rather than video recordings; p. 4, line 183). We also note that transcripts were stripped of any potentially identifying information on p. 5, lines 204-205.

- In the discussion I also miss a further analysis of the limitations of the study. For example, the non-segmentation of the participants is recognized but this may have some effect on the results. Or also, the interviews were by telephone, so a lot of non-verbal information is lost that could be interesting for the analysis.

We have addressed these points on pg. 12, lines 476-481.

Round 3

Reviewer 3 Report

I think the manuscript has been significantly improved, and attended all the suggestions.
As the only improvement, I suggest continuing by ordering better the presentation of the quotations of the participants from the interviews and focal groups, as I continue to perceive some disorder that could be improved.

Thank you very much and congratulations for the effort.

Author Response

Thank you for these additional comments and opportunity to revise the manuscript. We have gone through the results and re-organized them to try and clarify when results are from interviews versus when they are from focus groups.